# Effects of Muscular Strength Training on Oral Health and Quality of Life: Using Korean Panel Survey Data, a Cross-Sectional Study

**DOI:** 10.3390/healthcare11162250

**Published:** 2023-08-10

**Authors:** Kyeung-Ae Jang, Yu-Rin Kim

**Affiliations:** Department of Dental Hygiene, Silla University, Busan 46958, Republic of Korea; jka@silla.ac.kr

**Keywords:** exercise, muscles, oral health, quality of life

## Abstract

Objectives: The purpose of this study is to confirm the importance of muscular strength exercise by confirming the relationship between strength exercise, oral health, and quality of life. Methods: Using the 2019 and 2021 of the Korean National Health and Nutrition Examination Survey (KNHANES), 6535 people were selected as subjects. Complex sampling analysis was applied to all analyses; 2267 people were in the muscular strength training group (MSG), and 5841 people were in the non-muscular strength training group (NMSG). A multi-sample linear regression analysis was conducted to confirm the effect of muscular strength training on oral health and quality of life. Results: As a result of confirming the effect of muscular strength training on oral health status, problems with chewing decreased by 0.105, and problems with speaking decreased by 0.028 with MSG compared to NMSG. In addition, compared to NMSG, it was confirmed that MSG reduced chewing discomfort by 0.047, while self-perceived oral health improved by 0.0123. Finally, as a result of confirming the effect of muscular strength training on oral health and quality of life in Korean adults, there was a significant effect on quality of life despite adjusting for sociodemographic characteristics and oral-health-related factors (*p <* 0.05). Conclusions: In this study, the relationship between muscular strength training and quality of life was confirmed. Therefore, efforts should be made to make oral health management and muscular strength training a part of life in relation to quality of life.

## 1. Introduction

Recently, exercise has been gradually receiving attention as one of the ways to strengthen immunity, which can help prevent and treat chronic diseases. Lack of physical activity is a risk factor for chronic diseases, including cardiovascular diseases, cancer, diabetes, obesity, hypertension, bone and joint diseases, and depression [1]. Thus, active physical activity is important for individual and demographic health [2] and is necessary to address middle-aged adults’ health problems [3].

Among various physical activities, strength training acts as a strong stimulant for muscle protein synthesis and muscle cell growth [4]. It is also known to be effective in metabolic syndrome by continuously increasing muscle mass and strength, enhancing basal metabolic rate, and controlling body weight [5]. Strength training is essential for modern people as it effectively improves motor skills and prevents adult diseases and joint diseases. Nevertheless, the physical activity rate of Korean adults is 38.3% [6], and the strength exercise rate is 23.6% [7], which is very low. The low participation rate in strength training is probably because strength training itself is difficult and boring.

As people age, the quantity and quality of skeletal muscle decline, which is also observed in the oral and maxillofacial areas [8,9]. Orofacial generally refers to areas such as the lips, cheeks, and tongue, and the strength of these areas is called orofacial strength. Muscle strength of the cheeks and lips is effective for oral bolus containment and manipulation, and proper muscle tension of the cheeks and lips during chewing and swallowing prevents food from flowing into the oral vestibule between the gums and cheeks. Positive pressure is formed in the oral cavity, and the bolus can be swallowed efficiently when the tongue muscle strength is high [10]. Conversely, an imbalance in nutritional intake is possible due to discomfort in chewing if the oral muscle strength is weakened. This weakens the whole body’s health and deteriorates one’s chewing function, especially for those who are old [11]. In addition, when problems with pronunciation and swallowing functions occur, in severe cases, these can lead to dysphagia and aspiration pneumonia [12]. When oral muscle strength decreases, soft foods are selectively consumed, inhibiting the activation of masticatory muscles, leading to a vicious cycle in which oral muscle strength further decreases [13]. Therefore, oral muscle strength is included in systemic muscle strength, and loss of muscle strength can cause problems in oral health and general health, which is also related to the quality of life [11,12,13].

Groessl et al. [14] reported that senior citizens’s physical function and health-related quality of life improved when moderate physical activity was increased. Previous studies in other countries [15,16,17] also revealed that senior citizens with high levels of physical activity had a higher health-related quality of life than those who did not, and similar results were obtained in domestic studies [18,19,20] dealing with the association between the level of physical activity and health-related quality of life of the senior citizens in Korea. As such, although strength training is essential in improving quality of life, most of the studies on muscle strength and quality of life have only focused on senior citizens. Therefore, it is necessary to expand the research area to adults to confirm the effect of strength training on their oral health and quality of life.

Health-related quality of life means seeking more than economic security, and it comprehensively reflects all dimensions related to one’s well-being, including income, housing environment, and social support [21]. The European Quality of Life 5 Dimensions (EQ-5D) is widely used as a national survey tool for measuring the quality of life as it has the advantage of simple questions and a short survey time. However, since the EQ-5D was a tool developed in Europe, and there were differences between countries outside [22] and within Europe [23], some researchers contextualized the instrument. It is difficult to reflect the unique characteristics of each country even in domestic studies [24], so the Korea Disease Control and Prevention Agency (KDCA) developed the Korean Health-related Quality of Life Instrument with 8 Items (HINT-8) to measure more accurately the health-related quality of life according to Koreans’ characteristics [25]. As a result of conducting a survey using the HINT-8 questionnaire for Korean adults, 12.3% of the respondents answered that there was no problem in all areas, indicating that the ceiling effect was lower than that of the EQ-5D [26]. This means that HINT-8 can show the health-related quality of life of the public more precisely than EQ-5D. However, since HINT-8 was first introduced in the first year of the 8th KNHANES, it was not properly utilized due to the lack of follow-up research.

This study aims to use the data from the KNHANES in 2019 and 2021, when the HINT-8 survey was conducted, to identify the differences in oral-health-related factors and quality of life according to Korean adults’ strength training. Therefore, the hypotheses of this study are as follows:

**H0.** 
*Strength training may not be related to oral health and quality of life.*


**H1.** 
*Strength training may be related to oral health and quality of life.*


## 2. Materials and Methods

### 2.1. Study Design

In this study, the HINT-8 was used in 2019 in the KNHANES conducted by KDCA, followed by a two-year cycle survey using data from 2019 to 2021, excluding 2020. In 2019, the number of participants in at least one of the health surveys, checkup surveys, and nutrition surveys was 10,859, and the participation rate was 74.7%. In 2021, the number of participants in one or more areas was 9682, and the participation rate was 73.2%.

Among the 12,297 physical examination and nutrition survey participants, 7354 people between 19 and 65 years old were selected as the study’s respondents. In the health behavior survey (self-administered survey), 6535 people were chosen as study subjects after excluding missing values (Figure 1).

### 2.2. Setting

The KNHANES is a monitoring program on the health behavior of the people, the prevalence of chronic diseases, and the actual state of food and nutritional intake, which is conducted in accordance with Article 16 of the National Health Promotion Act and is government-designated statistics (Approval No. 117002) based on Article 17 of the Statistics Act. The KDCA provides only data that have been de-identified so that individuals cannot be estimated from survey data in compliance with the Personal Information Protection Act and Statistics Act. In addition, the Center for Disease Control and Prevention obtained consent from the subject to participate in research on human derivatives, and the study was conducted. The data used in this study were from the first and third years of the 8th survey. The review of research ethics was resumed after the collection of human-derived materials and the provision of raw data to third parties and was approved by the Institutional Review Board (IRB) (2018-01-03-C-A and 2018-01-03-3C-A).

### 2.3. Study Participants

In the physical activity category, the answers to the question ‘How many days did you do push-ups, sit-ups, dumbbells, weights, barbells, etc., in the last week?’ are not at all, 1–5 or more days, not applicable (children), and don’t know and no response. In this study, the non-muscular strength group (NMSG) answered not at all, while the muscular strength group responded 1 to 5 days or more.

### 2.4. Variables

#### 2.4.1. Demographic Characteristics

Gender, age, marital status, education level, income, economic activity status, presence or absence of a doctor’s diagnosis in the health questionnaire, alcohol consumption, and smoking were identified using the KNHANES’s health questionnaire.

The respondents’ age was divided into 19–35, 36–45, 46–55, and 56–64 years old. Marriage status was classified into married and single. The level of education was categorized into elementary school graduate or less, middle school graduate, high school graduate, and college graduate or higher, while income was classified by income quintile (household). The economic activity status was determined using the participants’ yes (the employed) or no (the unemployed, economically inactive population) responses. The presence or absence of a doctor’s diagnosis in the health questionnaire consisted of hypertension, dyslipidemia, stroke, myocardial infarction, angina pectoris, diabetes, and depression. Alcohol consumption was divided into never drank and drank, and smoking was divided into less than 100 cigarettes in a lifetime, more than 100 cigarettes in a lifetime, and never smoked.

#### 2.4.2. Oral Health Related Factors

Among oral-health-related factors, it was found that the respondents’ oral health behavior, such as whether they brushed their teeth during the previous day, included four oral care products—dental floss, interdental brush, mouth rinse, and toothbrush). In addition, four items—gum disease treatment, tooth decay treatment, root canal treatment, and preventive care—were the reasons for the respondents’ oral examination, scaling, and decision to visit a dental clinic in the past year. As for oral health status, the respondents complained about chewing discomfort, chewing problems, speaking problems, and self-recognized oral health status. For chewing and speaking problems, a high score means that there is no problem, and a low score means a good self-recognized oral health status.

#### 2.4.3. Health-Related Quality of Life Instrument with 8 Items (HINT-8)

The HINT-8 tool was developed by the KDCA in 2014. In the current study, this tool measured eight health-related items of the respondents’ past week’s activities. The sub-items consisted of climbing stairs, pain, energy, work, depression, memory, sleep, and happiness and were measured on a 4-point scale. A high score indicates a low health-related quality of life. The HINT-8 tool implies that a low score means a high quality of life-related to health [25,27].

### 2.5. Data Sources/Measurement

This study aimed to confirm the difference between strength exercise and oral health and quality of life and to confirm the relationship between strength exercise and quality of life. Oral-health-related factors, oral-health-related practices, and oral-related conditions were identified. In addition, health-related quality of life was checked, and the lower the score, the better the quality of life.

### 2.6. Bias

When we analyzed the association between strength training and quality of life, we presented Model 2, which corrected for sociodemographic characteristics to minimize bias. In addition, Model 3, which corrected oral-health-related factors, was presented to analyze the influencing factors.

### 2.7. Study Size

This study was analyzed using the KNHANES data provided by the KCDA. Therefore, it was analyzed using data from 2019 to 2021 when the variables for HINT-8 were investigated.

### 2.8. Quantitative Variables

HINT-8 is a 4-point scale, and lower scores mean higher health-related quality of life. Additionally, for chewing and speaking problems, the higher the score, the fewer problems there are, and the lower the score, the better the perceived oral health status.

### 2.9. Statistical Methods

Data analysis was performed using IBM SPSS version 21.0 (IBM Co., Armonk, NY, USA), and complex sampling analysis with stratification variables, cluster variables, and weights was applied in all analyses. A total of 8108 people were divided into 2267 people in MSG, a strength training group, and 5841 people in NMSG, a non-strength training group. A complex sample chi-square test was conducted to compare demographic characteristics and oral-health-related factors. Then, complex sample linear regression analysis was conducted to confirm the effect of strength training on the perceived oral health problems and HINT-8. In terms of the effect on health-related HINT-8, no adjustment was made in Model 1, demographic characteristics were adjusted in Model 2, and Model 3 was adjusted after adding oral health status to Model 2. Don’t know, non-applicable, and missing values were excluded in 8, 9, 88, and 99. The number of subjects in all tables was presented as an unweighted frequency, and the significance level of the statistical test (*p*-value) was 0.05.

## 3. Results

### 3.1. Demographic Characteristics According to Strength Training Status

Most MSG members are male and 19–35 years old, whereas the majority of NMSG members are female and are 36–45 years old. Regardless of strength training status, both groups had a high percentage of married people with higher income level, education level, and higher economic activity. Moreover, most participants are working, and they have drinking experience but do not have smoking experience. There was a significant difference between the two groups in all variables (*p* < 0.05) (Table 1).

### 3.2. Differences in Doctor Diagnosis According to Strength Training Status

Hypertension, dyslipidemia, myocardial infarction, angina pectoris, diabetes, and depression were the most common diagnoses in the NMSG. There were significant differences between NMSG and MSG’s hypertension and dyslipidemia diagnoses (*p* < 0.05) (Table 2).

### 3.3. Differences in Oral-health-related Factors According to Strength Training Status

In MSG, brushing teeth during the previous day, dental floss, interdental brush, mouth rinse, and toothbrush were all used more than those in NMSG. Then, MSG’s percentage of visits to dental clinics and scaling was higher than that of NMSG. There were significant differences in the groups’ use of dental floss, visits to dental clinics, and root canal treatment (*p* < 0.05) (Table 3).

### 3.4. The Effect of Strength Training Status on Oral Health Status

The effect of strength training on oral health was confirmed. Compared to NMSG, MSG had a score of 0.105 for no problems with chewing and 0.028 for no problems with speaking. Additionally, MSG was −0.047 for complaints of discomfort about chewing and −0.0123 for self-perceived oral health status. Thus, improvement was confirmed compared to the NMSG (*p* < 0.05) (Table 4).

### 3.5. Effects of Strength Training on Health-Related Quality of Life

As a result of confirming the effect of strength training on health-related quality of life, all items had a significant effect on health-related quality of life (*p* < 0.05). Model 2, which was adjusted for demographic characteristics, and Model 3, which was adjusted by adding oral-health-related factors to Model 2, showed significant effects in all items except for depression and sleep (*p* < 0.001) (Table 5).

## 4. Discussion

Muscular strength increases until age 30, and the current state of muscle strength can be maintained until 50. However, after 60, muscle strength decreases by 30% every 10 years [28,29]. This is due to a decrease in the number or size of muscle fibers, and the contractile force of the muscle reduces [30]. This decrease in muscle strength leads to decreased activity, leading to faster muscle and skeletal deterioration, which in turn results in muscle weakness and ultimately lowers the quality of life [31]. Therefore, this study investigated the relationship between adult Koreans’ oral-health-related factors and quality of life according to their strength training status using the KNHANES data in 2019 and 2021 and the HINT-8 questionnaire.

The study’s findings revealed that strength training was related to health-related quality of life, and Park [32] found that the group with strength training less than twice a week had a higher risk of health-related problems, unlike the group with more than twice-a-week strength training who do not experience such risks. Ahn et al. [33] supported the results of this study by reporting that the health-related quality of life of the group that performed strength training for 5 or more days per week was significantly higher than that of the group that exercised for 1 day or less. According to their study, the group that completed strength training for 1 day or less was 4.07 times more likely to have problems with exercise capacity than the group that completed strength training for 5 days or more per week. The probability of having problems with exercise capacity increased by 1.31 times as the number of strength training per week decreased by 1 day. In this study, MSG (strength training group) had the most significant effect on stair climbing and energy compared to NMSG (non-strength training group) among the sub-items of quality of life. These results are similar to the results of Kwon and Park [34], which confirmed that grip strength and strength of the lower extremity were improved with a 12-week strength training program as a practical effect of strength training.

In addition, the study illustrated that strength training improves muscle strength and endurance, which also helps in performing daily functional activities and reducing injury, disability, and mortality [35]. Previous studies support the study’s results that emphasize how strength training frequency and quality of life positively affect one’s work, reduce pain, promote happiness, and improve memory. In the case of memory, particularly, a study confirmed the change of work memory in the group [36] reported that aerobic exercise improves memory after 12 weeks of aerobic exercise improves memory, which is similar to the results of this study, but it is not consistent with the results in this study because the type of exercise is different.

In this study, sleep and depression were irrelevant to strength training, which is similar to a study’s findings where middle- or high-intensity strength training did not reduce healthy senior citizens’s depression [37]. However, it is a study concerned with senior citizens, while the present study focused on adults. In addition, based on the results of previous studies on the effect of physical activity on the reduction in depression, ACSM [38] reported that there is a solid basis for high-intensity strength training effective in treating depression. Moreover, ACSM [38] suggested that strength training [39,40] and aerobic exercise [41] effectively reduce senior citizens’s depression. Accordingly, the quality of life should be identified in various measures, including the time and duration of strength training for Korean adults.

The orofacial muscles play an important role in moving food to the base of the tongue and influencing the transition from the oral stage to the oropharyngeal stage. Among the orofacial muscles, the orbicularis oris, a muscle mainly involved in swallowing, closes the lips to prevent food from spilling out of the mouth, and the buccinator prevents food from flowing out of the mouth by closing the lips and flattening the cheeks as food moves across the teeth [42]. These functional orofacial muscles play an important role in swallowing, mastication, breathing, and speaking [10]. In this study, we confirmed the effect of strength training on oral health status. As a result, chewing problems, speaking problems, and chewing discomfort were lower in MSG than in NMSG. The subjective oral health condition was better. These results are similar to those of Kang [43], whose study presented that when grip strength is low, problems with chewing and speaking will be high, and subjective health perception is low.

Additionally, in oral health practice, the use of oral care products was high when the grip strength was high, like the results of this study. In this study, the frequency of dental floss use was high in MSG. Dental floss requires the removal of foreign substances and plaque in the oral cavity using the hand’s small muscles and repetitive motions, so it requires grip strength. Hence, muscle strength is related to oral hygiene habits. People visit dental clinics as a preventive measure and seek treatment for oral problems when strength training programs are not received. However, since no study directly confirmed the relationship between strength exercise and oral health, further research on various variables of oral health conditions considering the frequency and time of strength training is needed.

In terms of sociodemographic characteristics, the MSG members are male, young, married, have higher income and studied, are employed and non-smokers, and have drinking experience. Therefore, it is necessary to propose easily accessible muscle strength training considering the characteristics of the subjects vulnerable to strength training. In this study, most NMSG members have hypertension and dyslipidemia. Strength training not only improves physical function but also prevents and treats chronic diseases [5]. In addition, a decrease in skeletal muscle strength and muscle mass was reported as the number of circulatory inflammatory markers increased, suggesting that muscle strength is highly related to chronic diseases [44]. Thus, proper strength training is very important to improve systemic and oral health, including quality of life.

Although this study used data that can represent Koreans, it has some limitations. First, as the analysis was performed using cross-sectional survey data from 2019 to 2021, the causal relationship between strength training, oral health status, and quality of life could not be identified. Accordingly, the researcher recommends that future studies be conducted by establishing a cohort. Second, since the analysis was performed using secondary data, the strength training time was not obtained, so there is a limit to confirming the difference in the effects according to the exercise time. Therefore, further research is suggested to see the effects of reducing oral health problems and lowering quality of life according to the degree and time of strength training. Nevertheless, this study is meaningful in confirming the need for strength training to improve the quality of life of adults, unlike previous studies that were limited to specific subjects such as senior citizens or patients.

In addition, it is significant that strength exercise and oral health were confirmed together rather than a simple comparison of strength exercise and quality of life using national data. Therefore, this study is significant as it serves as a basis for developing easily accessible strength training promotion and exercise education programs, including push-ups, sit-ups, dumbbells, weights, and barbells.

## 5. Conclusions

This study analyzed data from Korean adults’ responses to the KNHANES. The muscular strength group’s strength training affected their oral health status and quality of life compared to the non-muscular strength group (non-strength training group). Therefore, it is essential to publicize the necessity of easily accessible strength training programs and develop policies that can help improve adults’ quality of life. In addition, experts need to develop and support effective and easily accessible strength exercise programs to enable continuous strength exercise.

## Figures and Tables

**Figure 1 healthcare-11-02250-f001:**
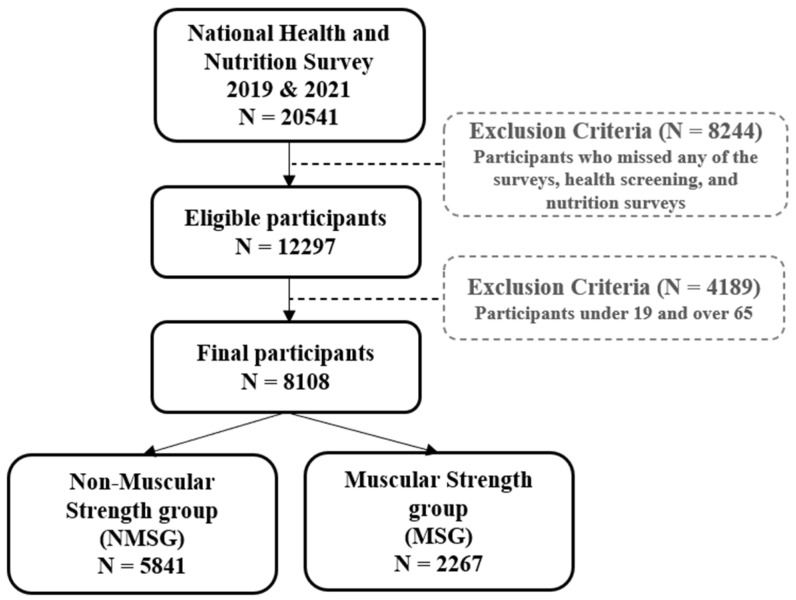
Study design.

**Table 1 healthcare-11-02250-t001:** General characteristics of subjects (weighted %).

Characteristics	Division	MSG	NMSG	*p* *
Gender	Male	1351 (65.5)	2249 (44.7)	<0.001
	Female	916 (34.5)	3592 (79.1)	
Age	19–35	769 (40.8)	1408 (29.4)	<0.001
	36–45	464 (19.4)	1422 (74.3)	
	46–55	476 (19.6)	1563 (26.3)	
	56–64	558 (20.1)	1448 (20.4)	
Marriage	Married	1523 (61.6)	4547 (73.7)	<0.001
	Single	744 (38.4)	1294 (26.3)	
Income level	1st quintile (lowest)	134 (5.3)	378 (5.7)	<0.001
	2nd quintile	246 (10.5)	859 (14.2)	
	3rd quintile	471 (21.1)	1350 (23.7)	
	4th quintile	615 (28.0)	1588 (27.8)	
	5th quintile (highest)	793 (35.0)	1645 (28.6)	
Education level	<Elementary school	356 (4.6)	61 (2.3)	<0.001
	Middle School	433 (6.3)	133 (4.5)	
	High School	2316 (40.1)	858 (39.1)	
	>College	2733 (49.0)	1215 (54.2)	
Economic activity	Employed person	15,579 (69.1)	4047 (70.0)	0.048
	Unemployed person	1793 (30.0)	710 (30.9)	
Drinking experience	Experience	2171 (30.2)	5465 (69.8)	0.002
	No experience	93 (3.8)	370 (5.7)	
Smoking experience	Less than 100 cigarettes in a lifetime	62 (3.2)	144 (2.9)	<0.001
	More than 100 cigarettes in a lifetime	1029 (46.8)	2074 (38.9)	
	Never smoked	1173 (50.0)	3617 (58.5)	

* By complex sample chi-square test, *p* < 0.05.

**Table 2 healthcare-11-02250-t002:** Differences in doctor’s diagnosis with and without strength exercises (weighted %).

Characteristics	MSG	NMSG	*p* *
Diagnosis of hypertension (yes, %)	281 (10.9)	904 (14.3)	<0.001
Diagnosis of Dyslipidemia (yes, %)	294 (11.0)	945 (14.8)	<0.001
Diagnosis of stroke (yes, %)	21 (1.0)	58 (1.0)	0.796
Diagnosis of myocardial infarction (yes, %)	15 (45.2)	23 (54.8)	0.086
Diagnosis of angina pectoris (yes, %)	13 (0.5)	49 (0.8)	0.163
Diagnosis of diabetes (yes, %)	123 (4.9)	375 (6.1)	0.080
Diagnosis of depression (yes, %)	92 (4.0)	287 (4.5)	0.338

* By complex sample chi-square test, *p* < 0.05.

**Table 3 healthcare-11-02250-t003:** Differences in oral-health-related factors according to strength exercise (weighted %).

Characteristics	MSG	NMSG	*p* *
Did you brush your teeth yesterday? (yes, %)	2257 (99.7)	5793 (99.3)	0.132
Did you use a floss yesterday? (yes, %)	822 (35.2)	1841 (31.9)	0.011
Did you use an interdental brush yesterday? (yes, %)	682 (26.7)	1562 (25.6)	0.347
Did you use gargle yesterday? (yes, %)	581 (31.4)	1322 (23.2)	0.109
Did you use an electric toothbrush yesterday? (yes, %)	167 (7.3)	402 (6.9)	0.575
Have you had regular checkups in the past year? (yes, %)	1040 (43.5)	2422 (67.2)	0.083
Have you been to the dentist in the past year? (yes, %)	1453 (61.8)	3422 (58.1)	0.007
Have you had an oral examination in the past year? (yes, %)	1372 (94.3)	3244 (94.8)	0.556
Have you had periodontal treatment in the past year? (yes, %)	246 (15.7)	673 (18.9)	0.010
Have you received simple tooth decay treatment in the past year? (yes, %)	373 (26.4)	956 (29.0)	0.116
Have you received root canal treatment in the past year? (yes, %)	258 (17.4)	713 (21.4)	0.009
Have you received oral preventive care in the past year? (yes, %)	52 (7.9)	142 (9.1)	0.367
Have you had tartar removed in the past year? (yes, %)	531 (76.1)	1150 (74.2)	0.416

* By complex sample chi-square test, *p* < 0.05.

**Table 4 healthcare-11-02250-t004:** Effects of strength training on oral health status.

Division	*β*	*t*	*p* *
No problems with chewing	0.105	5.567	<0.001
No problem with talking	0.028	2.495	0.013
Problems complaining of discomfort about chewing food	−0.047	−5.573	<0.001
Self-recognized oral health status	−0.123	−5.471	<0.001

* By complex sample linear regression analysis, *p* < 0.05, Reference category; NMSG. R2 = (1; 0.004, 2; 0.001, 3; 0.004, 4; 0.008).

**Table 5 healthcare-11-02250-t005:** Effects of strength training on quality of life (HINT-8).

Division	M1			M2			M3		
	*β*	*t*	*p* *	*β*	*t*	*p* *	*β*	*t*	*p* *
Climbing stairs	0.201	15.410	<0.001	0.152	11.225	<0.001	0.137	8.829	<0.001
Pain	0.114	6.544	<0.001	0.057	3.182	0.002	0.057	2.881	0.004
Energy	0.173	7.697	<0.001	0.146	6.213	<0.001	0.130	4.682	<0.001
To work	0.120	7.081	<0.001	0.098	5.723	<0.001	0.087	4.744	<0.001
Melancholy	0.068	3.824	<0.001	0.030	1.722	0.086	0.027	1.364	0.173
Memory	0.081	5.533	<0.001	0.046	3.021	0.003	0.035	2.069	0.039
Sleep	0.041	2.351	0.019	0.020	1.093	0.275	0.020	0.978	0.329
Happiness	0.096	4.541	<0.001	0.071	3.209	0.001	0.053	2.135	0.033
HINT-8 total	0.893	10.020	<0.001	0.621	6.850	<0.001	0.545	5.400	<0.001

* By complex sample linear regression analysis, *p* < 0.05, Reference category; NMSG. Model 1 was unadjusted; Model 2 was adjusted for demographic characteristics; Model 3 was adjusted for demographic characteristics and oral health-related factors. Model 1’s R2 = (1; 0.028, 2; 0.007, 3; 0.010, 4; 0.008, 5; 0.002, 6; 0.004, 7; 0.001, 8; 0.003, 9; 0.015). Model 2’s R2 = (1; 0.097, 2; 0.051, 3; 0.033, 4; 0.051, 5; 0.056, 6; 0.046, 7; 0.024, 8; 0.045 9; 0.082). Model 3’s R2 = (1; 0.110, 2; 0.075, 3; 0.052, 4; 0.078, 5; 0.075, 6; 0.073, 7; 0.041, 8; 0.066, 9; 0.133).

## Data Availability

The data presented in this study are available on request from the corresponding author.

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
