# Peer review of "Effects of Muscular Strength Training on Oral Health and Quality of Life: Using Korean Panel Survey Data, a Cross-Sectional Study"

_healthcare, 2023, doi:10.3390/healthcare11162250_

Round 1

Reviewer 1 Report

Effects of Muscular Strength Training on Oral Health and Quality of Life: Using Korean Panel Survey Data, A Cross-sectional Study

Overall I find it a timely, well-structured and interesting article. I consider that its publication may be necessary but small changes are needed, which I detail below:

ABSTRACT:

Improve it and you should make the study more understandable. Just by reading the abstract, the reader should know what the article is about, the methodology used, the results and the most important conclusion.

INTRODUCTION:

Clear and complete. I think it is correct.

The objective this study was  confirm the importance of muscular strength

exercise by confirming the relationship between strength exercise, oral health, and quality of life. From my point of view, I consider this study interesting because it relates the strength of the oral musculature to the quality of life of patients.

MATERIAL AND METHODS

The study should have the approval of an Ethics Committee. This should be pointed out and explained in the methodology or at the end of the article, but I believe that this fact is essential for its publication.

This study provides an important sample size, which makes its publication interesting.

RESULTS

Corrects

DISCUSSION:

-It should be improved by comparing the results obtained with other studies.

-I find the limitations of this study very well described and timely. I think it would be interesting to add the strengths of the study.

CONCLUSION

The conclusions are in line with the objectives.

REFERENCES

The bibliographic references are sufficient, although they could be expanded and, as mentioned above, the results of the present study could be discussed further with another study.

TABLES AND FIGURES

The tables and figures seem to me to be correct

Author Response

Reviewer 1

Comments and Suggestions for Authors

Effects of Muscular Strength Training on Oral Health and Quality of Life: Using Korean Panel Survey Data, A Cross-sectional Study

Overall I find it a timely, well-structured and interesting article. I consider that its publication may be necessary but small changes are needed, which I detail below:

ABSTRACT:

Improve it and you should make the study more understandable. Just by reading the abstract, the reader should know what the article is about, the methodology used, the results and the most important conclusion.

 Answer: Thank you very much for your views. We have included only the important points in the abstract to make it easier for the reader to understand.

INTRODUCTION:

Clear and complete. I think it is correct.

The objective this study was  confirm the importance of muscular strength

exercise by confirming the relationship between strength exercise, oral health, and quality of life. From my point of view, I consider this study interesting because it relates the strength of the oral musculature to the quality of life of patients.

Answer: Thank you very much for your views.

MATERIAL AND METHODS

The study should have the approval of an Ethics Committee. This should be pointed out and explained in the methodology or at the end of the article, but I believe that this fact is essential for its publication.

This study provides an important sample size, which makes its publication interesting.

 Answer: This study analyzed national data. Therefore, we have received a number approved by the country, and its contents are inserted in the text.

[[The KNHANES is a monitoring program on the health behavior of the people, the prevalence of chronic diseases, and the actual state of food and nutritional intake, which is conducted in accordance with Article 16 of the National Health Promotion Act and is government-designated statistics (Approval No. 117002) based on Article 17 of the Statistics Act. The Korea Centers for Disease Control and Prevention provides only data that has been de-identified so that individuals cannot be estimated from survey data in compliance with the Personal Information Protection Act and Statistics Act. In addition, the Center for Disease Control and Prevention obtained consent from the subject to participate and research on human derivatives, and the study was conducted. The data used in this study were from the first and third years of the 8th survey. The review of research ethics was resumed after the collection of human-derived materials and the provision of raw data to third parties and was approved by the Institutional Review Board (IRB) (2018-01-03-C-A and 2018-01-03-3C-A).]]

RESULTS

Corrects

Answer: Thank you very much for your views.

DISCUSSION:

-It should be improved by comparing the results obtained with other studies.

Answer: We compared strength training and quality of life with other studies [32-40]. In addition, the results of strength training and oral health were compared with reference [43]. It was difficult to find studies that confirmed the relationship between strength training and oral health, and it was difficult to compare the results of our study simply because it was difficult to find results consistent with this study. So hope for your understanding.

-I find the limitations of this study very well described and timely. I think it would be interesting to add the strengths of the study.

 Answer: Thank you very much for your views. We have added the strengths of our study as per your opinion.

CONCLUSION

The conclusions are in line with the objectives.

 Answer: Thank you very much for your views.

REFERENCES

The bibliographic references are sufficient, although they could be expanded and, as mentioned above, the results of the present study could be discussed further with another study.

  Answer: Thank you very much for your views.

TABLES AND FIGURES

The tables and figures seem to me to be correct

Answer: Thank you very much for your views.

Reviewer 2 Report

Comments for Manuscript ID: healthcare-2542128

General comments

Authors demonstrated that strength exercise had effect on quality of life and oral health using national survey of Korea. Although the results is potentially interesting, there are some issues to address for acceptance to the Healthcare.

Abstract

L18 What is NSG?

Introduction

L41-42 “Continuous participation is…” Please indicate references for this statement.

L45 “the strength of these areas is called orofacial strength.

L48 “into the sulcus between the gums and cheeks.” Does the wors “sulcus” mean oral vestibule?

L49 The word “food“ should be changed to “bolus” or “food of bolus.”

L58 Please describe the logical basis and references for the statement “loss of muscle strength can cause problems in oral health and general health.”

L60 Negative terms such as “the elderly” must be avoided due to their ageist or stereotypical connotations. Similar in other sentences.

Methods

SHS, GAD Please indicate in full spelling.

2.3. Study participants

Do participants include individuals currently serving in the military?

L192 Results Does “oral health status” mean self-recognized oral health status?

Table 4 Why did not analyze by models adjusted for HINT-8 and demographic characteristics similar as Table 5?

L222-225 The interpretation regarding regression coefficients seems to be incorrect. Regression coefficients represent the contribution rate of the input independent variables.

L296 The word “decreased” seems to be incorrect, since the study is cross-sectional study.

Discussion

L300-309 While authors’ interpretations may also be possible, the interpretation that there is a correlation between oral hygiene habits and exercise habits seems more appropriate.

Authors should be more mindful of using specialized terminology overall.

Author Response

Reviewer 2

Comments and Suggestions for Authors

Comments for Manuscript ID: healthcare-2542128

General comments

Authors demonstrated that strength exercise had effect on quality of life and oral health using national survey of Korea. Although the results is potentially interesting, there are some issues to address for acceptance to the Healthcare.

Abstract

L18 What is NSG?

 Answer: Thank you very much for your views. We corrected the misspelled words.

Introduction

L41-42 “Continuous participation is…” Please indicate references for this statement.

Answer: As the content is the judgment of the author, references are not provided. We have modified the text to make it easier for readers to understand.

L45 “the strength of these areas is called orofacial strength.

Answer: We cannot understand your review. What needs to be corrected on line 45?

L48 “into the sulcus between the gums and cheeks.” Does the wors “sulcus” mean oral vestibule?

Answer: Thanks for your views. We have corrected the terminology.

L49 The word “food” should be changed to “bolus” or “food of bolus.”

Answer: Thanks for your views. We have corrected the terminology.

L58 Please describe the logical basis and references for the statement “loss of muscle strength can cause problems in oral health and general health.”

Answer: We have inserted references to this text.

L60 Negative terms such as “the elderly” must be avoided due to their ageist or stereotypical connotations. Similar in other sentences.

Answer: We changed “the elderly” to “senior citizens” throughout the text.

Methods

SHS, GAD Please indicate in full spelling.

Answer: We have corrected the content. thank you.

2.3. Study participants

Do participants include individuals currently serving in the military?

Answer: Those in military service were excluded from the subjects of the Korea National Health and Nutrition Examination Survey. This is because data collection is limited. Therefore, those in military service are not subject to data collection from the beginning.

L192 Results Does “oral health status” mean self-recognized oral health status?

Answer: We set the variable content for oral health status in [2.4.2. Oral Health related factors] section.

Table 4 Why did not analyze by models adjusted for HINT-8 and demographic characteristics similar as Table 5?

Answer: This study tried to confirm Table 3 (strength exercise and oral health related factors) and Table 4 (strength exercise and oral health status) together. Table 5 is the final analysis of meaningful results. Therefore, Tables 3 and 4 are analyzed in the same context, and Table 5 is the final analysis we want to talk about. In conclusion, we applied the analysis method differently. We appreciate your understanding.

L222-225 The interpretation regarding regression coefficients seems to be incorrect. Regression coefficients represent the contribution rate of the input independent variables.

Answer: We modified the description of Table 4. thank you.

L296 The word “decreased” seems to be incorrect, since the study is cross-sectional study.

Answer: We have modified the content according to your opinion. thank you.

Discussion

L300-309 While authors’ interpretations may also be possible, the interpretation that there is a correlation between oral hygiene habits and exercise habits seems more appropriate.

Answer: We have modified the terminology according to your views and appropriate places in the flow of the content. Our research focuses on strength training, oral health-related factors (including oral health care behavior and oral health status), and quality of life, rather than oral hygiene habits and exercise habits. So we appreciate your understanding.

Comments on the Quality of English Language

Authors should be more mindful of using specialized terminology overall.

Answer: We received English proofreading from native speakers. thank you.

Reviewer 3 Report

I have read and reviewed the manuscript entitled: Effects of Muscular Strength Training on Oral Health and Quality of Life: Using Korean Panel Survey Data, A Cross-sectional Study".

The authors should respond to the following questions:

Throughout the manuscript the authors should change the word "quality of life" to the term "oral-health-related quality of life".

TITLE:

- Is it necessary in the title to state "Using Korean Panel Survey Data"?

ABSTRACT:

- In the abstract the authors indicate "background" but make no introduction.

- The authors should add the keyword "oral-health-related quality of life".

INTRODUCTION:

- Why was only the period from 2019 to 2021 considered?

- The introduction is correct and well justified bibliographically.

MATERIALS AND METHODS:

- Which author(s) conducted the data analysis?

- Has the tool for analysing oral-health-related quality of life been validated before, and why have the authors not used another analysis tool? Authors should adequately justify that point in the manuscript.

RESULTS:

- Considering that the sample studied is not homogeneous with respect to age and gender, have the authors assessed the influence of age and gender on the results of this study? This point is very important.

- How did the authors determine the variables analysed in table 2?

CONCLUSIONS:

- The authors describe the following sentence in the conclusions "In addition, the active support of workplaces and local governments in the creation of productive and effective programmes, such as physical exercises led by health or fitness experts, is recommended to enable continuous strength training", do the results of the present study report this above conclusion?

Author Response

Reviewer 3

Comments and Suggestions for Authors

I have read and reviewed the manuscript entitled: Effects of Muscular Strength Training on Oral Health and Quality of Life: Using Korean Panel Survey Data, A Cross-sectional Study".

The authors should respond to the following questions:

Throughout the manuscript the authors should change the word "quality of life" to the term "oral-health-related quality of life".

Answer: As noted in the text, we did not examine oral health-related quality of life. We investigated quality of life (HINT-8). Thank you for your consideration.

TITLE:

- Is it necessary in the title to state "Using Korean Panel Survey Data"?

Answer: When country data is used, it is judged that it is common to display it in the title. thank you.

ABSTRACT:

- In the abstract the authors indicate "background" but make no introduction.

- The authors should add the keyword "oral-health-related quality of life".

Answer: We modified [Objectives:] to fit the content, and since oral quality of life was not investigated, the content was not included in the keywords. Thank you for your consideration.

INTRODUCTION:

- Why was only the period from 2019 to 2021 considered?

- The introduction is correct and well justified bibliographically.

 Answer: We reviewed the survey year [1. Introduction] and [2.1. Study Design] in the first sentence. Please note.

[1. Introduction; This study aims to use the data from the KNHANES in 2019 and 2021, when the HINT-8 survey was conducted, to identify the differences in oral health-related factors and quality of life according to Korean adults’ strength training.]

[2.1. Study Design; In this study, the HINT-8 was used in 2019 in the annual National Health and Nutrition Survey conducted by KDCA, followed by a two-year cycle survey, using data from 2019 and 2021, excluding 2020.]

MATERIALS AND METHODS:

- Which author(s) conducted the data analysis?

- Has the tool for analysing oral-health-related quality of life been validated before, and why have the authors not used another analysis tool? Authors should adequately justify that point in the manuscript.

 Answer: Thank you very much for your views. We added the person who analyzed the data to [Author Contributions]. In addition, we analyzed the data surveyed by the quality of life (HINT-8) scale tailored to Koreans, not the quality of life related to oral health. The reasons for not using another quality-of-life measure, the EQ-5D, are few enough in the introduction.

[The European Quality of Life 5 Dimensions (EQ-5D) is widely used as a national survey tool for measuring the quality of life as it has the advantage of simple questions and a short survey time. However, since the EQ-5D was a tool developed in Europe, and there were differences between countries outside [22] and within Europe [23], some researchers contextualized the instrument. It is difficult to reflect the unique characteristics of each country even in domestic studies [24], so the Korea Disease Control and Prevention Agency (KDCA) developed the Korean Health-related Quality of Life Instrument with 8 Items (HINT-8) to measure more accurately the health-related quality of life according to Koreans’ characteristics [25]. As a result of conducting a survey using the HINT-8 questionnaire for Korean adults, 12.3% of the respondents answered that there was no problem in all areas, indicating that the ceiling effect was lower than that of the EQ-5D [26]. This means that HINT-8 can show the health-related quality of life of the public more precisely than EQ-5D.]

Since this data was surveyed for Koreans, the data was analyzed with a quality of life scale suitable for Koreans. We appreciate your understanding.

RESULTS:

- Considering that the sample studied is not homogeneous with respect to age and gender, have the authors assessed the influence of age and gender on the results of this study? This point is very important.

- How did the authors determine the variables analysed in table 2?

 Answer: As we finally presented Table 5, Model 2 corrected demographic and sociological characteristics, and Model 3 corrected by adding oral health status to Model 2. Therefore, it is judged that there is no difficulty in interpreting the results of this study. In addition, [2.4.1. Demographic Characteristics] explains the presence or absence of a medical diagnosis corresponding to Table 2.

[The presence or absence of a doctor's diagnosis in the health questionnaire consisted of hypertension, dyslipidemia, stroke, myocardial infarction, angina pectoris, diabetes, and depression.]

CONCLUSIONS:

- The authors describe the following sentence in the conclusions "In addition, the active support of workplaces and local governments in the creation of productive and effective programmes, such as physical exercises led by health or fitness experts, is recommended to enable continuous strength training", do the results of the present study report this above conclusion?

Answer: We modified the last sentence of section [5. Conclusions]. thank you.

Round 2

Reviewer 2 Report

Authors addressed all issues which the reviewer indicated. There are no additional comments for the manuscript.